# Construction of a Diagnostic m^7^G Regulator-Mediated Scoring Model for Identifying the Characteristics and Immune Landscapes of Osteoarthritis

**DOI:** 10.3390/biom13030539

**Published:** 2023-03-16

**Authors:** Liang Hao, Xiliang Shang, Yang Wu, Jun Chen, Shiyi Chen

**Affiliations:** Huashan Hospital, Fudan University, Shanghai 200040, China

**Keywords:** osteoarthritis, 7-methylguanosine, biomarkers, scoring model

## Abstract

With the increasingly serious burden of osteoarthritis (OA) on modern society, it is urgent to propose novel diagnostic biomarkers and differentiation models for OA. 7-methylguanosine (m^7^G), as one of the most common base modification forms in post transcriptional regulation, through which the seventh position N of guanine (G) of messenger RNA is modified by methyl under the action of methyltransferase; it has been found that it plays a crucial role in different diseases. Therefore, we explored the relationship between OA and m^7^G. Based on the expression level of 18 m^7^G-related regulators, we identified nine significant regulators. Then, via a series of methods of machine learning, such as support vector machine recursive feature elimination, random forest and lasso-cox regression analysis, a total of four significant regulators were further identified (DCP2, EIF4E2, LARP1 and SNUPN). Additionally, according to the expression level of the above four regulators, two different m^7^G-related clusters were divided via consensus cluster analysis. Furthermore, via immune infiltration, differential expression analysis and enrichment analysis, we explored the characteristic of the above two different clusters. An m^7^G-related scoring model was constructed via the PCA algorithm. Meanwhile, there was a different immune status and correlation for immune checkpoint inhibitors between the above two clusters. The expression difference of the above four regulators was verified via real-time quantitative polymerase chain reaction. Overall, a total of four biomarkers were identified and two different m^7^G-related subsets of OA with different immune microenvironment were obtained. Meanwhile, the construction of m^7^G-related Scoring model may provide some new strategies and insights for the therapy and diagnosis of OA patients.

## 1. Introduction

Being unpredictable and one of the most common chronic degenerative joint diseases, OA (OA) has a prevalence that increases with age. OA causes significant pain and disability [1,2,3]. Furthermore, inflammation or fibrosis of the infrapatellar fat pad is present in patients with OA, which is one of the well-established risk factors for the development of the pain caused by OA [4,5]. Moreover, pathological changes, including subchondral osteosclerosis, synovitis, fibrosis, and cartilage degeneration are closely associated with OA [6,7,8,9]. OA has a multifaceted etiology; and is caused by a combination of immune response, chronic inflammation, trauma, and biomechanical processes [10,11,12]. Congenital joint abnormalities, trauma, stress injury, obesity, sex, age, and knee gap narrowing [13] are all complicatedly associated with the development of OA [14].OA can have serious physical, emotional, and economic consequences; moreover, it is becoming an evolving public health issue that negatively affects the daily lives and quality of life of people [15]. With an increasing prevalence, overall, the age-standardized prevalence of OA increased by 7.5% in Northern Europe between 1990 and 2015, with an annual increase of 43% [16]. Clinical symptoms and imaging findings are required for making the standard diagnosis of OA; however, often, when these symptoms appear, patients have already reached the late stage of the disease [17]. Moreover, patients with OA show relatively severe symptoms and have poor treatment results because no current drug treatment has been found to reverse the progression of OA in the long term [15]. Therefore, it is even more important to look for novel diagnostic modalities that can help diagnose OA as early as possible, facilitating the timely treatment of patients.

RNA modifications were not known as the “epitranscriptome” until 2015. Emerging studies on the function of these modifications have shown significant implications for human pathology [18]. Adenosine methylation is present in mRNAs and non-coding RNAs, such as circular RNA (circRNAs), microRNAs (miRNAs), and long-stranded non-coding RNAs (lncRNAs), which adjust their biogenesis and function [19]. The RNA methylation types can be divided into various modification types: m^6^A, m^5^C, m^7^G, and so on; of which m^6^A and m^7^G are the two most common types [20,21,22]. However, so far, more extensive studies on the association between m^6^A and OA are available, whereas there are few studies on m^7^G [23,24]. Moreover, there is a prevalence of m^7^G RNA modifications within mRNA, and their conservation, regulation, and dynamics as well as their roles in translational control have been shown. Modifications of the m^7^G cap are widely seen in OA-related mRNA [25]; they play an important role in the efficient translation, splicing, and stability of related mRNA and also affect the synthesis of related proteins. Modifications of m^7^G are also observed in OA-related mRNA and help in enabling the translation of OA-related mRNA. Thus, m^7^G, as a transcriptional marker, is important for protein translation and can be used as the basis for making diagnostic models for OA.

Herein, we have analyzed numerous publicly available microarray datasets. Using differential analysis and algorithmic screening, we have acquired the most critical genes of m^7^G-regulators for forming an intersection. We have identified four significant regulators, including DCP2, EIF4E2, SNUPN, and LARP1, in combination with machine learning. Based on the expression of these four regulators, we have divided all the OA samples into m^7^G-related clusters. Next, we performed principal component analysis (PCA) to calculate the m^7^G score in the above two clusters. Then, we performed differential expression analysis, enrichment analysis, and immune infiltration to explore the characteristics of the two clusters. Finally, after the intersection of the differentially expressed genes (DEGs) between the above two clusters and DEGs between the normal samples (NM) and OA samples, we constructed a diagnostic model using LASSO Cox regression. Furthermore, we verified the abovementioned four regulators with the use of real-time quantitative polymerase chain reaction (RT-qPCR).

## 2. Materials and Methods

### 2.1. Data Acquisition and Processing

By retrieving the keyword “OA” from the Gene Expression Omnibus database, datasets containing the synovial tissue samples of normal people and patients with OA were obtained [26]. Moreover, the standards for screening our datasets are as follows: (1) Homo sapiens Expression Profiling by array, (2) synovial tissue of OA from joint synovial biopsies, (3) datasets containing complete information about the samples, (4) one biopsy sample per subject was analyzed without replicates. The detailed information regarding the datasets used in our study is listed in Table 1. Then, “inSilicoMerging” [27] was used to merge and the “limma” package (v3.42.2) in R software [28] was used to remove the batch effect of these three data sets; finally, a data set containing the synovial tissue samples of 26 patients with OA and 20 normal people was obtained. Furthermore, 24 m^7^G-regulators collected in the gene sets GOMF_RNA_7_METHYLGUANOSINE_CAP_BINDING, GOMF_M7G_5_PPPN_DIPHOSPHATASE_ACTIVITY, and GOMF_RNA_CAP_BINDING, were summarized, wherein only 18 regulators were annotated in our data sets [29]. In this study, the association between the expression of these 18 genes and diseases as well as that between m^7^G-regulators self-expression was studied. The results were visualized using heat maps. The Wilcoxon signed-rank test was performed to select some significant regulators.

### 2.2. Enrichment Analysis

Gene ontology (GO) analysis is a common method to annotate gene products and the functions of genes, including cellular component, biological pathway, and molecular function [30]. The Kyoto Encyclopedia of Genes and Genomes (KEGG) is a useful database for the systematic analysis of gene functions and associated high-level genomic functional information [31]. We used “clusterProfiler” (version 3.14.3) and R software packages “org.Hs.eg.db” (version 3.1.0) for the GO and KEGG pathway analysis. The maximum size gene sets were set to 5000 genes and the minimum to 5; the analysis results with a *p*-value of <0.05 were considered significant. ClueGO is an important plug-in of Cytoscape, which can be used for GO enrichment analysis. It was also used for enrichment analysis in our study [32].

### 2.3. Construction and Verification of Prediction Model

To more accurately screen the m^7^G-regulators associated with the occurrence of OA, significant m^7^G-regulators and OA were extracted using the support vector machine recursive feature elimination (SVM-RFE) algorithm and random forest algorithm (RF) [33], and the two analysis results were intersected using the Venn diagram. Finally, to determine the best prognostic characteristic regulators, the LASSO Cox regression analysis was performed on the basis of the abovementioned results [34]. Then, cluster analysis was performed on the finally screened feature genes using the “ConsensusClusterPlus” package; accordingly, the samples were divided into two categories. PCA was used for extracting PC1 and PC2 to form signature scores. Afterward, the above scores were applied to construct the m^7^G score: **m^7^G score = Σ (PC1_i_ + PC2_i_).**

### 2.4. Immune Infiltration

The genes significantly associated with 28 immune cell types from the literature were collected (Appendix A). Then, the expression of these immune genes were linked with the distribution of 28 types of immune cells using the single-sample gene set enrichment analysis method [35]; combined with our m^7^G regulators; we finally analyzed the association of 28 immune cells with different m^7^G-related clusters and immune cells with m^7^G characteristic genes.

### 2.5. Construction of Diagnostic Model of OA

To better explore the characteristic of our two m^7^G-related clusters, differential expression analysis was performed to assess the DEGs between the two m^7^G-related clusters and those between the NM and OA samples with the cutoff criteria of |log2FC| > 1 (*p* < 0.05). After the intersection of the two different types of DEGs, a diagnostic model was constructed using the overlapped DEGs via the LASSO Cox regression. The diagnostic score was as follows: **Diagnostic Score = ∑i Coefficients_i_ * Expression level of signature_i_**.

### 2.6. Collection of Our Synovial Samples

In our study, synovial tissue samples of OA were obtained from patients undergoing surgery due to knee OA (n = 15). The normal synovial samples were from patients who underwent surgery for meniscus laceration of the knee joint caused by trauma (n = 15).

### 2.7. RT-qPCR 

Trizol (Invitrogen, Waltham, MA, USA) reagent was used to extract total RNA from the synovial tissue samples of normal people and patients with OA; the Prime Script TMRT kit (Takara, RR047A) was used to reverse transcribe to obtain cDNA. Finally, the SYBR Premix Ex Taq II Kit (Takara, Japan) was used for PCR amplification according to the manufacturer’s instructions. And we used Bio-Rad (CFX96) of the UK for RT-qPCR. The primer sequences are listed in Appendix A.

### 2.8. Statistical Analysis

R software (version 4.2.1) and its related software packages were used to process and analyze data (*p* < 0.05). We used the Wilcoxon signed-rank test to assess the significance of difference between the two groups. And a *t*-test was used for the analysis of the result of RT-qPCR. Afterward, Sangerbox was used to visualize the results of the receiver operating characteristic curves and Wilcoxon signed-rank test; the “RMS” package in R software was used to visualize Nomograms.

## 3. Results

### 3.1. Identification of Significant m^7^G-Regulators in OA

We calculated the Spearman correlation coefficient among these 18 regulators, wherein several regulators demonstrated significant correlation on the basis of the expression levels of 18 m^7^G-regulators in our datasets which were merged with the three datasets from GEO (Figure 1A). Meanwhile, the interaction association of these 18 regulators was revealed by constructing a protein-protein interaction network (Figure 1B). Then, we used the Wilcoxon signed-rank test to identify the significant regulators in our training set. Thus, nine significant regulators (*p* < 0.05) were obtained, including DCP2, IFIT5, EIF4E2, NUDT11, NUDT3, LARP1, SNUPN, LSM1, and CYFIP1 (Figure 1C). The heat map shows the expression level of these nine significant regulators (Figure 1D).

### 3.2. Enrichment Analysis for Our Significant m^7^G-Regulators

The ClueGO plug-in in Cytoscape was first used to perform enrichment analysis (*p* < 0.05) for comprehensively exploring the function of the above nine significant m^7^G-regulators in OA. Thus, the term with the largest proportion was “m^7^G(5′)pppN diphosphatase activity” (53.85%); furthermore, the rest of the terms revealed that our significant m^7^G-regulators were almost involved in the pathways of RNA metabolism and translation progress (Figure 2A). Meanwhile, the MCODE plug-in was used to extract the important clusters of the ClueGO results. The clusters with a high score are shown in Figure 2B–D; the cluster with the highest score was the “m^7^G(5′)pppN diphosphatase activity” pathway, which was consistent with the biological function of these regulators.

Moreover, the GO, KEGG, and Reactome analyses were performed to ensure the preciseness of our research (*p* < 0.05) (Figure 2E–I). Correspondingly, almost all results of different methods of enrichment analysis revealed that these regulators focused on RNA metabolism. Interestingly, pathways associated with the immune system and other pathways, including the viral myocarditis, interferon signaling, and HIF-1 signaling pathways, were found, indicating that these m^7^G-regulators played a significant role in RNA modification as well as immune and other fields.

### 3.3. Selection of Significant m^7^G-Regulators via Machine Learning

In line with the above analysis, we explored the important and key role of OA by using several methods of machine learning to further identify some significant regulators in OA. First, SVM-RFE was performed to evaluate the diagnostic effectiveness of these regulators. Thus, seven regulators were obtained. Meanwhile, another method, RF, was used to calculate the importance of these regulators (Figure 3A,B). With a score of >2, six regulators were selected (Figure 3C,D). Then, we intersected the results of SVM-RFE and RF, through which four regulators (EIF4E2, DCP2, SNUPN, and LARP1) were considered the final crucial regulators (Figure 3E).

Furthermore, the LASSO Cox regression was performed to verify the diagnostic effectiveness of these four regulators (Figure 3F,G). Thus, all four regulators were regarded as significantly diagnostic signatures.

Moreover, based on the expression matrix of GSE32317, the nomogram was further exhibited to reveal the efficiency of the above four m^7^G-regulators in distinguishing early- and end-stage OA, and the calibration curve revealed the accuracy of our model (Figure 3H,I).

### 3.4. Identification of Two Different m^7^G-Related Clusters

According to the expression levels of the four key regulators selected via machine learning, we divided our OA samples in the training set into two m^7^G-related clusters with the most appropriate K value (K = 2) via consensus cluster analysis (Figure 4A–C). Furthermore, the PCA diagram revealed a significant difference between clusters A and B (Figure 4D). Meanwhile, all four regulators showed a significant statistical difference between m^7^G-related clusters A and B (Figure 4E,F). Based on the PCA algorithm, an m^7^G score module was calculated to distinguish the above two clusters (*p* < 0.05), which was higher in m^7^G-related cluster B and lower in m^7^G-related cluster A (Figure 4G).

### 3.5. GSEA, Immune Infiltration, and Immune Checkpoint Characteristics in m^7^G–Related Clusters

To better describe the characteristics and functions in the abovementioned m^7^G-related clusters, the GSEA analysis was performed. We identified three pathways with a *p*-value of <0.05, including TGF_BETA_SIGNALING, ALLOGRAFT_REJECTION, and ESTROGEN_RESPONSE_EARLY, which indicated that the m^7^G score was mainly associated with the metabolism and immune system (Figure 5A). Therefore, we performed the immune infiltration analysis and the mantel test to demonstrate the association between the four significant regulators and the infiltration score of the 28 immune cells. Interestingly, a stronger association was noted in cluster A, indicating that the lower m^7^G score suggested prominently elevated infiltration of immune cells in patients with OA (Figure 5B,C). 

Furthermore, the Pearson correlation coefficient between the expressions of a series of immune checkpoint-related genes in the above two clusters and the m^7^G score was calculated for more comprehensively exploring the immune signature between these two clusters. For the immune checkpoint inhibitors, the m^7^G score in m^7^G-related cluster A was positively associated with most of the inhibitors (Figure 5D). On the contrary, the m^7^G score in m^7^G-related cluster B was negatively correlated with most inhibitors (Figure 5E).

### 3.6. Exploration of Difference between the above Two Clusters and Construction of a Diagnostic Model

To further emphasize the significance of the m^7^G-related clusters, differential expression analysis was performed with |log2FC| > 1 (*p* < 0.05) as the cutoff. Thus, 113 DEGs were identified, including 46 upregulated and 67 downregulated, which were visualized using the volcano and heat maps (Figure 6A,B). Moreover, the DEGs between NM and OA and between the above two clusters intersected. Finally, eight overlapped DEGs were obtained (CRYBB1, N6AMT1, SNORA21, HAUS2, P2RX3, RRN3P1, CC2D1A, and FKBP5), which were considered the candidate factors extracted for the LASSO Cox regression (Figure 6C,D). We regarded lambda-min: 0.0469 as the optimal value after running the 10-fold cross-validation. Thus, five factors were selected to construct our diagnostic model for OA: Diagnostic value = (−0.00806730919977491 × SNORA21) + (−0.00181179794976226 × HAUS2) + (−0.0134814157908044 × CC2D1A) + (−0.00123114088948981 × FKBP5) + (0.00983579963789009 × N6AMT1) (Figure 6E,F). Furthermore, all samples were randomly divided into two different subsets with a ratio of 1:1: a verification set and a training set. According to the above formula of the diagnostic score, the Wilcoxon signed-rank test was performed to explore the statistical difference between the NM and OA samples. Finally, the diagnostic score of both sets demonstrated a significant difference (*p* < 0.05) (Figure 6G,I). In addition, the area under the receiver operating characteristic curve of the diagnostic model in the verification and training set was 85.4701 and 93.0070, respectively, which further indicated the excellent effectiveness of our diagnostic model (Figure 6H,J). Moreover, to further verify the accuracy of the diagnostic model, another external dataset GSE12021 was selected, wherein a significant difference was observed in the diagnostic value between NM and OA samples (*p* < 0.05) (Figure 6K). Meanwhile, in GSE12021, the area under the receiver operating characteristic curve of the diagnostic model was 97.7778, demonstrating the accuracy of our model (Figure 6L).

### 3.7. Validation of the Four Significant m^7^G-Regulators in the Synovial Tissue of Patients with OA 

To explore the abovementioned four significant m^7^G-regulators (DCP2, EIF4E2, SNUPN, and LARP1) in OA, the synovial tissues of normal people and patients with OA were collected from the Second Affiliated Hospital of Nanchang University and the demographic data of the patients included in our study is listed in Table 2. Then, the expression difference in the RNA level was verified using qRT-PCR. Thus, except for LARP1, the other three regulators exhibited a significant high-expression level in OA, which was consistent with our above analysis (Figure 7A–D).

## 4. Discussion

OA is usually assumed to be caused by non-inflammatory factors; that is, a series of mechanical stresses that destroys the cartilage. However, recently, some associated inflammatory factors have also been shown to contribute to OA development, allowing inflammatory cells to infiltrate the synovium [36]. Moreover, inflammatory cytokines play an essential role in the progression of OA by stimulating the production of matrix metalloproteinases and thus increasing matrix degradation [37]. An increasing number of studies have focused on the effects of nucleic acid site changes on the cell function and even body activities, wherein RNA modification plays a critical role. This can be seen in the methylation of m^7^G, which is located in the inner part of tRNA and rRNA and plays a significant role in coordinating numerous functions during the mRNA lifecycle. This is largely accredited to the protein factors that particularly bind to the cap structures, the cap-binding complex in the nucleus, and eIF4E in the cytoplasm. m^7^G is accountable for mRNA processing and nuclear export in the nucleus and is needed for effective pre-mRNA splicing. In vivo, mRNA interacts with protein factors throughout its life cycle and also plays a role in transcription termination and exosome degradation [38]. In the present study, we have briefly discussed the association between m^7^G and OA. Several research gaps still exist despite a growing interest in this field. Thus, we have attempted to create a novel prospect for the clinical diagnosis of OA.

In this study, three datasets were downloaded from the Gene Expression Omnibus database, namely GSE55235, GSE55457, and GSE55584. After unified treatment, 18 regulatory factors associated with m^7^G were identified and analyzed from the data of 26 patients with OA and 20 normal people. Nine statistically significant m^7^G regulators were obtained; their association was strong enough. Second, a protein-protein interaction network was constructed to enrich and analyze these regulators using GO and KEGG. The results showed that these genes were mainly involved in no MTG-related dephosphorylation, RNA metabolism, RNA modification, and other processes and were enriched in hypoxia-related GFR signaling, insulin metabolism-related, and virus-related pathways. Next, two machine learning algorithms (RF and SVM-RFE) were used to further screen the nine genes, obtain the intersection, and further obtain four key genes, EIF4E2, DCP2, SNUPN, and LARP1. Based on these four genes, the diagnostic effectiveness of these four genes was further verified using the LASSO Cox regression. Furthermore, based on the expression levels of the abovementioned four hub genes, the 26 OA samples in the training set were divided into two different m^7^G-related clusters; the PCA algorithm was used to further calculate the m^7^G score to differentiate the two subtypes. Immune infiltration analysis demonstrated that cluster A was more closely associated with the immune system. Moreover, to better exhibit the characteristic of our m^7^G-related clusters, a diagnostic model was constructed for calculating the diagnostic score using the differential expression analysis and LASSO Cox regression. Finally, the differential expression of the abovementioned four genes in the synovial tissues of patients with OA was verified using an external validation set and RT-qPCR. Although unilaterally LARP1 exhibited a non-statistically significant difference between NM and OA in RT-qPCR, the m^7^G-related Score to distinguish two different m^7^G-related clusters of OA exhibited an accurate efficiency and the diagnostic model constructed via these four m^7^G-regulators played an extraordinary role in the diagnosis of OA. 

EIF4E2 is a protein-coding gene located in the p-body and belongs to part of the mRNA cap-binding active complex [39,40]. In the early stage of initiation, EIF4E2 recognizes and binds m^7^G mRNA cap, activates ubiquitin protein ligase-binding activity, and participates in miRNA-mediated translational inhibition. Contrary to EIF4E, EIF4E2 is unable to bind EIF4G (EIF4G1, EIF4G2, or EIF4G3), signifying that it assembles EIF4F by competing with EIF4E and blocking it in the cap region [41]. EEIF4E2-related diseases include casket-Siris syndrome 2, melanoma, cancer, viruses, and so on. The related pathways include the PI3K-Akt signaling and innate immune system pathways. RNA-binding and ubiquitin protein ligase binding are the GO annotations associated with this gene. Shaohong Chen et al. proposed that TNRC6 competes with EIF4E1 to recruit EIF4E2 for targeting mRNA, thus blocking translation initiation. Moreover, EIF4E2 mainly inhibits the expression of genes at the translational level but does not significantly affect the level of coding mRNA [42]. Moreover, Mir-29b is associated with EIF4E2. However, Mir-29b can silence premature AID expression in naive B cells, thus reducing the probability of inappropriate and potentially dangerous deamination activity [43]. Cadherin-22, a cell-cell adhesion molecule, is upregulated by promoter eIF4E2-mediated mTORC1-independent translational control during hypoxia; the novel function of cadherin-22 acts as a hypoxia-specific cell surface molecule and is involved in cancer cell migration, invasion, and adhesion [44]. However, no previous studies have noted and discussed the association between EIF4E2 and OA or the role of EIF4E2 in OA, which may become a potential topic for studies in the future.

The protein encoded by DCP2 is a key component of the mRNA uncoating complex required for mRNA degradation. It removes the 7-methylguanine cap structure from mRNA and then degrades from the 5′ end. The involved pathways are an unfolded protein response and the regulation of activated PAK-2P34 by proteasome-mediated degradation. The GO annotations associated with this gene include RNA binding and manganese ion binding. Moreover, T cell intracellular antigen-1 (TIA-1)-induced transformation silencing promotes the decay of selected mRNAs, and TIA-1-mediated decay is inhibited by small interfering RNAs targeting the 5′-3′ (e.g., DCP2) or 3′-5′ (e.g., exosomal component Rrp46) decay pathway, suggesting that TIA-1 sensitizes mRNA to both major decay pathways [45]. Interestingly, there is no study on the association between DCP2 and OA. This is also the novelty of our study, and it is worthy of more detailed research.

SNUPN is also a protein-coding gene, acts as a U snRNP-specific nuclear import adapter, and participates in the trimethylguanosine cap-dependent nuclear import of U snRNPs. SNUPN is associated with chronic lymphocytic leukemia, cancer, and so on. The related pathways include the translocation of pre-mRNA containing introns and SLBP-independent mature mRNA. The GO annotations associated with this gene include outdated protein transporter activity and RNA cap binding. Moreover, XPO1 binding to various proteins is mediated by the recognition of leucine-rich nuclear export signals at the N-terminus of SNUPN, thus transporting proteins. Overexpression or dysfunction of XPO1 has been reported in different cancers [46]. As the role of SNUPN in OA has not been explored by previous studies, our study is the first one to demonstrate a link between the two. To understand the role of SNUPN in OA in detail, further deeper research is necessary.

LARP1 is a class of RNA-binding proteins that regulates the translation of specific target mRNAs downstream of the mTORC1 complex and plays a role in growth signaling and nutrient availability while regulating cell growth and proliferation [47]. The diseases associated with LARP1 are dengue virus and hepatocellular carcinoma. The pathways involved include disease and SARS-CoV-2 infection. The GO annotations associated with this gene include RNA binding and translation initiation factor binding. Furthermore, RNMT selectively regulates the LARP1 target (TOP mRNA in the terminal polypyrimidine tract) expression. Increased ribosome abundance leads to the upregulation of RNMT for coordinating mRNA capping and processing and increasing translational capacity during T-cell activation [48]. Meanwhile, the association between LARP1 and OA is still not studied. Our study is the first one to show that LARP1 may have an impact on the pathogenesis of OA, but it still needs further exploration.

Furthermore, up to now, the diagnosis of OA has become clearer and accurate, which is based on X-rays and clinical symptoms [17,49,50]. However, with OA development, the joint pain caused by OA and its effect on the daily life and exercise capacity of patients becomes more and more severe. Thus, early and timely diagnosis of OA is urgent and necessary. In our study, the nomogram constructed based on the GSE32317 further revealed that our four regulators (DCP2, EIF4E2, SNUPN, and LARP1) also showed excellent accuracy to distinguish early- and end-stage OA, which provided effective insight to the early diagnosis of OA.

In summary, in this study, the collected data sets were unified and merged into a whole for analysis to avoid homogeneity. The number of samples used was also significant. Simultaneously, we used two different machine learning methods to avoid the one-sidedness of screening methods and make the results more convincing. Furthermore, the scoring model of m^7^G was constructed for the first time, which is progressive. However, not enough synovial tissue samples were collected. Moreover, the role of our four m^7^G-regulators in the occurrence and development of OA was not explored with detailed experiments, which needs to be explored in the future. Moreover, the five-gene diagnostic model obtained via our m^7^G-related cluster was verified only via external and internal datasets without experiments; thus, its diagnostic accuracy needs to be further validated. In conclusion, we constructed an m^7^G-related scoring model, which can significantly differentiate patients with OA, and correlated it with different statuses of the immune microenvironment, based on which we constructed a diagnostic model to diagnose patients with OA.

## Figures and Tables

**Figure 1 biomolecules-13-00539-f001:**
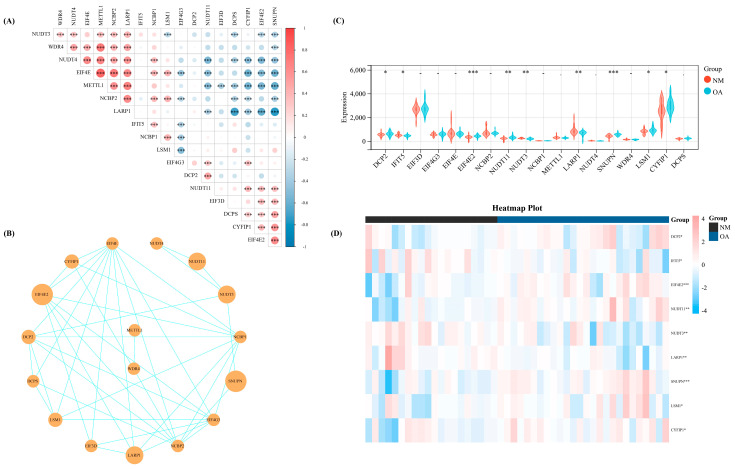
The basic landscape of m^7^G-regulators in OA. (**A**) The correlation heap map of the expression level of the 18 m^7^G-regulators. (**B**) The PPI network of the 18 m^7^G-regulators. (**C**) Th expression difference of the 18 m^7^G-regulators between NM and OA group. (**D**) The heap map of the nine significant regulators between NM and OA. -, *p* > 0.05; *, *p* < 0.05; **, *p* < 0.01; ***, *p* < 0.001. NM, normal; OA, OA.

**Figure 2 biomolecules-13-00539-f002:**
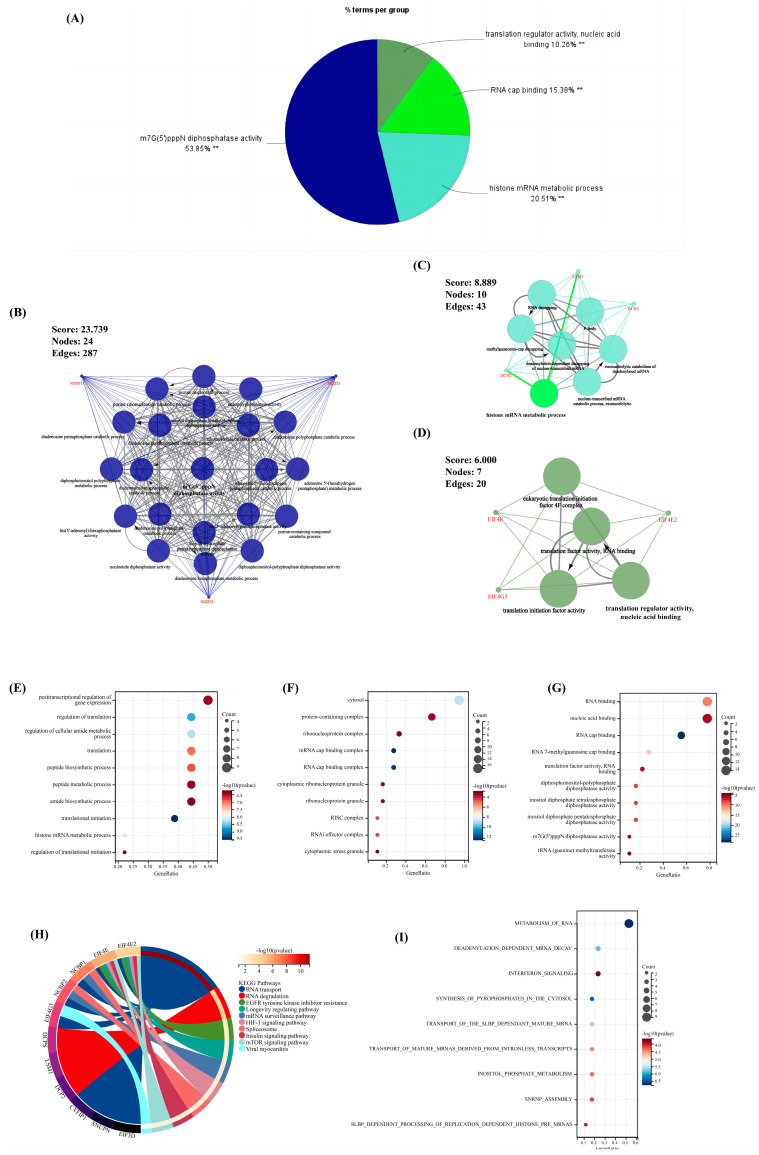
The results of enrichment analysis for the 18 m^7^G-regulators. (**A**) The pie chart of the results for ClueGO enrichment analysis. (**B**–**D**) Three significant clusters identified via MCODE plug-in for the results of ClueGO enrichment analysis. (**E**–**G**) The bubble chart of the results for GO enrichment analysis, (**E**) biological pathway (BP), (**F**) cellular component (CC), (**G**) molecule function (MF). (**H**,**I**) The result of the (**H**) KEGG and (**I**) Reactome enrichment analysis. **, *p* < 0.05.

**Figure 3 biomolecules-13-00539-f003:**
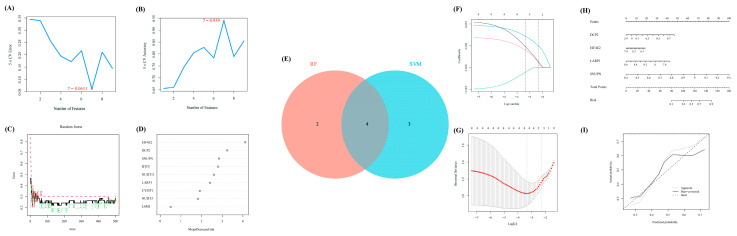
The further selection of crucial m^7^G-regulators for OA via machine learning. (**A**,**B**) The results of SVM-RFE identified seven significant features. (**C**,**D**) The results of RF identified six significant features. (**E**) The Venn diagram to extract the overlapped features between RF and SVM-RFE. (**F**,**G**) The results of lasso-cox regression analysis. (**H**) The nomogram of the four significant regulators to distinguish early- and end-stage of OA confirmed via machine learning. (**I**) The calibration curve of our nomogram.

**Figure 4 biomolecules-13-00539-f004:**
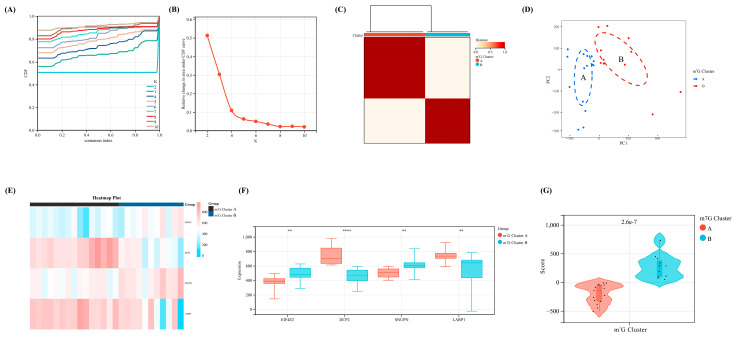
The identification of m^7^G-related clusters and construction of m^7^G scoring model. (**A**) The cumulative distribution curve of the result for consensus cluster analysis. (**B**) The area under the curve of the result for consensus cluster analysis. (**C**) The heat map of two different m^7^G-related clusters. (**D**) The PCA map of the above two different clusters. (**E**) The heap map of our four crucial regulators revealed the expression level between two clusters. (**F**) The expression difference of the four regulators between two clusters. (**G**) The difference of m^7^G score between two clusters. **, *p* < 0.01; ****, *p* < 0.0001.

**Figure 5 biomolecules-13-00539-f005:**
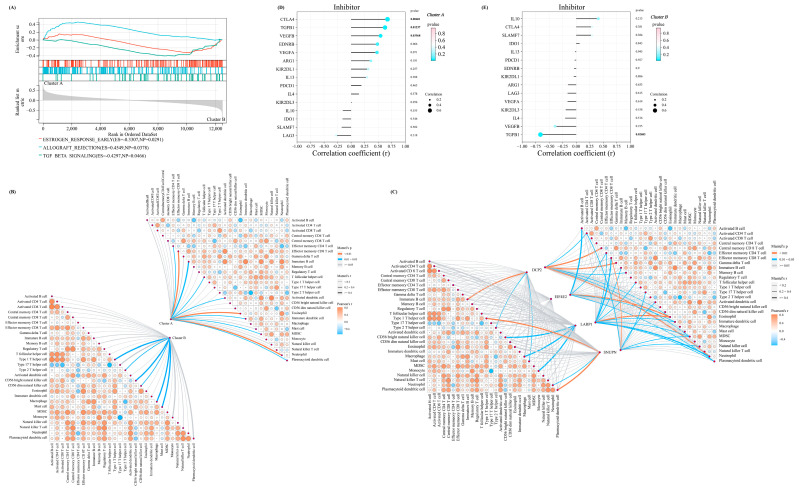
The distinctive characteristic of the two m^7^G-related clusters. (**A**) The results of the GSEA enrichment analysis in two clusters. (**B**) The mantel correlation heat map between the m^7^G score and 28 different kinds of immune cells in two different clusters. (**C**) The mantel correlation heat map between the expression level of four significant regulators and 28 different kinds of immune cells in two different clusters. (**D**,**E**) The lollipop chart revealed the results of Pearson correlation analysis between m^7^G score and the expression level of several immune checkpoint inhibitors in (**D**) cluster A and (**E**) cluster B.

**Figure 6 biomolecules-13-00539-f006:**
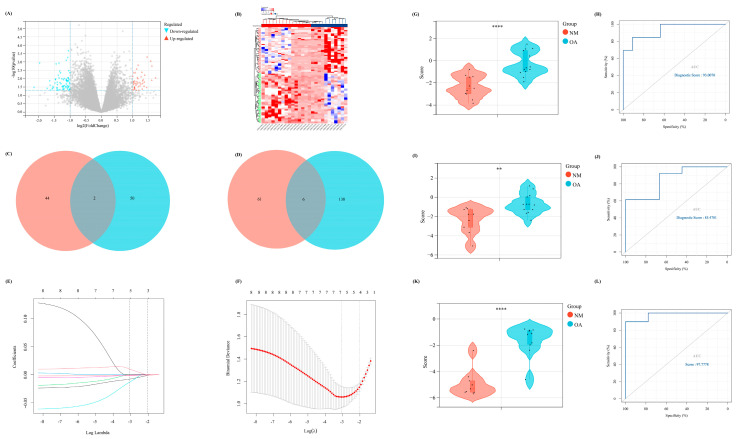
The construction of a diagnostic model related to m^7^G. (**A**) The volcano map of the DEGs between two different m^7^G clusters. (**B**) The heat map revealed the expression level of DEGs in two clusters. (**C**,**D**) The Venn diagram to intersect the overlapped (**C**) up-regulated and (**D**) down-regulated DEGs between the DEGs in two different clusters and DEGs between NM and OA group. (**E**,**F**) The results of lasso-cox regression analysis. (**G**,**I**,**K**) The difference of diagnostic score between NM and OA in our (**G**) training set, (**I**) internal verification set, and (**K**) external verification set. (**H**,**J**,**L**) The ROC curve of the diagnostic score in our (**H**) training set, (**J**) internal verification set, and (**L**) external verification set.

**Figure 7 biomolecules-13-00539-f007:**
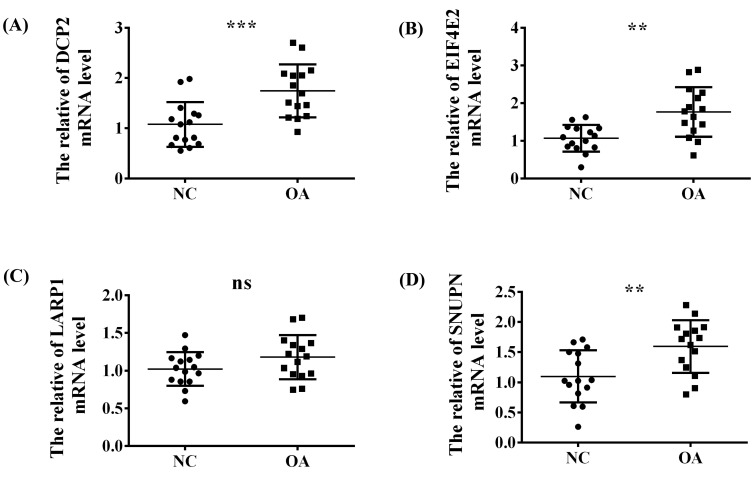
The validation experiment and *t*-test used to verify the expression trend of the four significant regulators (**A**) DCP2, (**B**) EIF4E2, (**C**) LARP1 and (**D**) SNUPN between the NC and OA group in the synovial tissue of our OA patients via RT-qPCR. ns, *p* > 0.05; **, *p* < 0.01; ***, *p* < 0.001.

**Table 1 biomolecules-13-00539-t001:** The information of datasets included in our research. OA, osteoarthritis; NM, normal.

Accession	Platform	Samples		Tissue
		NM	OA	
GSE55235	GPL96	10	10	synovium
GSE55457	GPL96	10	10	synovium
GSE55584	GPL96	0	6	synovium
GSE12021	GPL96	9	10	synovium
GSE32317	GPL570	Early OA	Late OA	synovium
		10	9	

**Table 2 biomolecules-13-00539-t002:** The demographic data of patients included in our study.

OA Samples				Normal Samples			
Age	Gender	Height (cm)	Weight (kg)	Age	Gender	Height (cm)	Weight (kg)
75	Female	154	60	43	Male	156	60
66	Male	168	66	53	Male	158	61
60	Female	155	55	53	Male	172	86
65	Male	163	70	68	Female	150	50
75	Female	150	50	54	Female	157	56
58	Female	158	65	55	Female	165	80
90	Male	160	61	54	Female	155	48
64	Female	163	63	57	Female	158	64
69	Female	155	58	56	Female	156	60
71	Female	150	55	55	Male	177	75
73	Male	178	90	47	Male	160	65
70	Female	160	45	60	Female	153	50
53	Male	171	71	61	Male	185	73
69	Female	150	50	54	Female	156	56
53	Female	150	46	53	Male	171	75

## Data Availability

The original data used in this project can be downloaded in the public database GEO (https://www.ncbi.nlm.nih.gov/geo/ accessed on 31 August 2022).

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
