# Peer review of "Construction of a Diagnostic m7G Regulator-Mediated Scoring Model for Identifying the Characteristics and Immune Landscapes of Osteoarthritis"

_biomolecules, 2023, doi:10.3390/biom13030539_

Round 1
Reviewer 1 Report
Dr. Liang Hao & Dr.Shiyi Chen,
it is a brilliant and exciting manuscript. Considering the novelty of the field I do consider that it is a prime manuscript. Most of my comments are directed to grammatics, cohesion, language precision, and citations.
Congratulations for the team effort.
My comments:
Text needs a severe upgrade in English grammatics and cohesion. This includes the abstract that does focus too much on methodology. As an example, the background for m7G is absent. Needs more internal structure… where do results start. The conclusion needs to be clear concise & precise. Avoid overgeneralizations such as “…we provided some new insights for the therapy… and so on…”
All statements need a clear citation. Example lines 29-31 statement is not supported by bibliography. Another example would be lines 47-63. Multiple statements with no citation. If it is an observation, suggestion… the right wording is needed. The whole manuscript needs to be revised.
All figures are misplaced. Figures should come just after their first citation
Legends need to be extended stating the used statistic system. The whole figure should be understandable standalone. Figure 2B, C, and D are difficult to read. Please adjust the text labeling to make it readable.
Author Response
Dear Reviewer, Thank you for your efforts to revise our manuscript. We have refined the full text of the manuscript with the help of professional institutions. In addition, misreferences have been corrected and we have added some novel and necessary citations. At the same time, we adjusted the size and order of the figures. We apologize for our mistakes and omissions. Thank you again for your contribution to our manuscript.
Reviewer 2 Report
Although the manuscript is interesting, there are several points that needs to be addressed by the authors.
Lines 29-31: meniscal degeneration and inflammation/fibrosis of the infrapatellar fat pad should be added. References should also be added.
Lines 31-32: the authors cited ref 2, which not appropriate. It should be checked and correct.
Lines 41-44, 59-61, 496-497: references should be added.
Line 54: m6A should be corrected.
The authors should report the figures near where they are cited.
Line 98: the three datasets should be reported and described. What criteria did the authors use in the dataset’s selection?
Lines 209-210: Inclusion/exclusion criteria of patient’s enrolment should be added. How did the authors obtain synovial samples from normal people? What did these persons have?
Lines 211-214: did the authors treat cDNA with DNAse? What kind of real time PCR instrument did the authors use?
In table S2 there is no mention about the housekeeping gene used. It should be added.
Figure 1A is too small.
Figure 1: NM should be defined.
Figures letter should be placed under the figures between round brackets.
Line 238: the authors write “training set”. Could the authors better explain what they mean? It is unclear to me if the authors use the three datasets mentioned in the methods to identify the Significant m7G-regulators in OA (section 3.1).
Lines 306-310: the authors propose a nomograph for the diagnosis of OA based on EIF4E2, DCP2, SNUPN, and LARP1. This part is questionable. Actually, there are no problems in the diagnosis of OA, which is based on X-rays and clinical symptoms. A big issue is early OA diagnosis rather than OA diagnosis. What would be the advantages of using this proposed nomogram?
Lines 415-416: it would be better to use different datasets as verification set.
Section 3.7: a table reporting the demographic data of the patients vs normal subjects should be added.
Figure 7: statistical test used should be indicated. Figure c should be LARP1 but it is reported “the relative of CXCL8 mRNA level” in the y axis. The authors should check the entire figure.
Lines 430-431: it seems that the validation of LARP1 failed. This is a big issue. It should be reported at lines 488-490.
Lines 491-509: are there studies on EIF4E2 and OA?
Lines 510-519: are there studies on DCP2 and OA?
Lines 520-528: are there studies on SNUPN and OA?
Lines 529-538: are there studies on LARP1 and OA?
Discussion should be improved. There is no discussion about these EIF4E2, DCP2, SNUPN, and LARP1 and OA. The nomogram should be discussed.
Limitations of the study should be discussed.
In general, it is suggested to avoid the use of light colors in the figures for the text, because of the difficulties in reading it. Moreover, it would be nice if the authors could enlarge the figures. Figures 5A, B and C cannot be appreciated.
Ethical approval number and date should be indicated.
It is not reported if the authors collect the informed consent from the humans involved in this study.
Abbreviations should be defined at first mention and used consistently throughout the manuscript (for example OA).
Author Response
Reviewer 2
Although the manuscript is interesting, there are several points that needs to be addressed by the authors.
Lines 29-31: meniscal degeneration and inflammation/fibrosis of the infrapatellar fat pad should be added. References should also be added.
Respond: Thank you for your remind, we have added in lines 34-37
Lines 31-32: the authors cited ref 2, which not appropriate. It should be checked and correct.
Respond: We are sorry for our mistakes and have corrected it in line 38
Lines 41-44, 59-61, 496-497: references should be added.
Respond: We apologized for our negligence and have added in lines 50-52, 59-64, 372-373
Line 54: m6A should be corrected.
Respond: We have corrected it in line 63
The authors should report the figures near where they are cited.
Respond: We have adjusted the position of our figures. Thank you for your careful advance.
Line 98: the three datasets should be reported and described. What criteria did the authors use in the dataset’s selection?
Respond: Thank you for your remind. We have added the criteria to select our datasets in lines 95-99 and the detailed information of our datasets in Table 1
Lines 209-210: Inclusion/exclusion criteria of patient’s enrolment should be added. How did the authors obtain synovial samples from normal people? What did these persons have?
Respond: Thank you for your remind. We have added the criteria in Section 2.6
Lines 211-214: did the authors treat cDNA with DNAse? What kind of real time PCR instrument did the authors use?
Respond: We have treated cDNA with DNAse and the PCR instrument used in our experiment is form the company: Bio-Rad (CFX96) of UK.
In table S2 there is no mention about the housekeeping gene used. It should be added.
Respond: Thank you for your remind. We have added it in Table S2
Figure 1A is too small.
Figure 1: NM should be defined.
Figures letter should be placed under the figures between round brackets.
Respond: We have added and adjusted the size and order of our figures
Line 238: the authors write “training set”. Could the authors better explain what they mean? It is unclear to me if the authors use the three datasets mentioned in the methods to identify the Significant m7G-regulators in OA (section 3.1).
Respond: The training set is the set merged with the three datasets from GEO.
Lines 306-310: the authors propose a nomograph for the diagnosis of OA based on EIF4E2, DCP2, SNUPN, and LARP1. This part is questionable. Actually, there are no problems in the diagnosis of OA, which is based on X-rays and clinical symptoms. A big issue is early OA diagnosis rather than OA diagnosis. What would be the advantages of using this proposed nomogram?
Respond: We have revised the original set used to draw our norogram. And the model constructed via morogram to distinguish early- and end-stage OA
Lines 415-416: it would be better to use different datasets as verification set.
Respond: We have added an external dataset GSE12021 as another verification set.
Section 3.7: a table reporting the demographic data of the patients vs normal subjects should be added.
Respond: We have added the demographic data of patients included in our study in Table 2.
Figure 7: statistical test used should be indicated. Figure c should be LARP1 but it is reported “the relative of CXCL8 mRNA level” in the y axis. The authors should check the entire figure.
Respond: We are sorry for our mistakes and have corrected in Figure 7.
Lines 430-431: it seems that the validation of LARP1 failed. This is a big issue. It should be reported at lines 488-490.
Respond: We have clarified and discussed this issue in lines 367-371.
Lines 491-509: are there studies on EIF4E2 and OA?
Lines 510-519: are there studies on DCP2 and OA?
Lines 520-528: are there studies on SNUPN and OA?
Lines 529-538: are there studies on LARP1 and OA?
Respond: There has never been any research on the relationship between the above four genes (DCP2, EIF4E2, SNUPN, LARP1) and OA
Discussion should be improved. There is no discussion about these EIF4E2, DCP2, SNUPN, and LARP1 and OA. The nomogram should be discussed.
Limitations of the study should be discussed.
Respond: We have added the limitations of our research in lines 430-438.
In general, it is suggested to avoid the use of light colors in the figures for the text, because of the difficulties in reading it. Moreover, it would be nice if the authors could enlarge the figures. Figures 5A, B and C cannot be appreciated.
Respond: We are sincerely sorry for our mistakes and negligence
Ethical approval number and date should be indicated.
It is not reported if the authors collect the informed consent from the humans involved in this study.
Respond: We have added the Institutional Review Board Statement and Informed Consent Statement in lines 467-471.
Abbreviations should be defined at first mention and used consistently throughout the manuscript (for example OA).
Respond: Thank you for your help. We have added it.
Round 2
Reviewer 2 Report
The manuscript improved after the revision. However, I have still some comments for the authors.
Lines 34-37: this part is confusing. Inflammation/fibrosis of the infrapatellar fat pad can be present in patients after anterior cruciate ligament reconstruction but importantly, inflammation/fibrosis of the infrapatellar fat pad is present in patients with OA. This should be specified.
Table 1: “NM” should be defined.
The authors replied to my question that they used Bio-Rad (CFX96) of UK for real time PCR. It should be added in the text.
In this version, the authors clarified that the “normal” synovial membrane samples used for the validation was taken from patients with meniscal tears. It is well known that synovial membrane of patients with meniscal tears is often inflamed and fibrotic (DOI: 10.3390/ijms23073903). This point should be reported and discussed.
Author Response
Dear Reviewer 2,
Thank you for your tireless efforts and valuable comments in the publish of our manuscript. All the revisions for your comment were fonts marked with red and our point-to-point responses to your comments are as follows:
Point 1. Lines 34-37: this part is confusing. Inflammation/fibrosis of the infrapatellar fat pad can be present in patients after anterior cruciate ligament reconstruction but importantly, inflammation/fibrosis of the infrapatellar fat pad is present in patients with OA. This should be specified.
Response 1: Thank you for your kindly remind. We have revised our previous confusing state in lines 34-37 into “Furthermore, inflammation or fibrosis of the infrapatellar fat pad is present in patients with OA, which is one of the well-established risk factors for the development of the pain caused by OA”.
Point 2. Table 1: “NM” should be defined.
Response 2: We are sincerely sorry for our carelessness and have added the definition of “NM” in the legend of Table 1 (Lines 104-105).
Point 3. The authors replied to my question that they used Bio-Rad (CFX96) of UK for real time PCR. It should be added in the text.
Response 3: Thank you for pointing out our negligence carefully and we have added it in line 155.
Point 4. In this version, the authors clarified that the “normal” synovial membrane samples used for the validation was taken from patients with meniscal tears. It is well known that synovial membrane of patients with meniscal tears is often inflamed and fibrotic (DOI: 10.3390/ijms23073903). This point should be reported and discussed.
Response 4: Thank you for your careful and thoughtful consideration. As we all know and according to the reference, the inflammation and fibrosis of synovial membrane often occurred in patients with meniscal tears. However, patients with meniscal tears which were included in our research suffer fresh damage, thus there was not enough time for synovium to be stimulated by inflammation and fibrosis. Therefore, the synovial membrane of our patients with meniscal tears can be regarded as the healthy and normal synovial tissue.
Yours sincerely,
Liang Hao.